# RainDiff: End-to-end Precipitation Nowcasting Via Token-wise Attention Diffusion

## Abstract

Precipitation nowcasting, predicting future radar echo sequences from current observations, is a critical yet challenging task due to the inherently chaotic and tightly coupled spatio-temporal dynamics of the atmosphere. While recent advances in diffusion-based models attempt to capture both large-scale motion and fine-grained stochastic variability, they often suffer from scalability issues: latent-space approaches require a separately trained autoencoder, adding complexity and limiting generalization, while pixel-space approaches are computationally intensive and often omit attention mechanisms, reducing their ability to model long-range spatio-temporal dependencies. To address these limitations, we propose a Token-wise Attention integrated into not only the U-Net diffusion model but also the spatio-temporal encoder that dynamically captures multi-scale spatial interactions and temporal evolution. Unlike prior approaches, our method natively integrates attention into the architecture without incurring the high resource cost typical of pixel-space diffusion, thereby eliminating the need for separate latent modules. Our extensive experiments and visual evaluations across diverse datasets demonstrate that the proposed method significantly outperforms state-of-the-art approaches, yielding superior local fidelity, generalization, and robustness in complex precipitation forecasting scenarios. Our code will be publicly released.

## 1 Introduction

Predicting when and where rain will fall over the next few minutes to hours, known as precipitation nowcasting, remains one of the most pressing challenges in weather forecasting (Ravuri et al., 2021; Veillette et al., 2020). The goal is to predict a sequence of future radar echoes conditioned on recent observations. Traditional approaches rely on numerical weather prediction (NWP), which explicitly models atmospheric dynamics through partial differential equations (PDEs) (Bauer et al., 2015). While physically grounded, NWP methods are computationally expensive and slow to update, limiting their use for the rapid, iterative forecasts required in nowcasting (Bi et al., 2023).

Recent advances in deep learning have enabled data-driven alternatives that bypass explicit PDE solvers. Deterministic nowcasting models have demonstrated a strong ability to capture the large-scale advection of precipitation fields, but they typically suffer from oversmoothing effects, especially at longer lead times (extending to several hours), resulting in underestimation of atmospheric intensity and loss of fine-scale spatial detail (Shi et al., 2015; Ravuri et al., 2021). To mitigate this, probabilistic generative approaches leveraging generative adversarial networks (GANs) or diffusion models have been introduced to generate more realistic and accurate radar fields (Zhang et al., 2023; Gao et al., 2023). However, these models often inflate the effective degrees of freedom by treating the entire spatio-temporal field as stochastic, which introduces excessive randomness and reduces the positional accuracy of rainfall structures (Yu et al., 2024).

To reconcile these trade-offs, hybrid architectures have emerged that benefit from both deterministic and probabilistic paradigms. These methods decompose weather evolution into (i) a globally coherent deterministic component to capture large-scale dynamics, and (ii) a localized stochastic refinement to model fine-grained variability (Yu et al., 2024; Gong et al., 2024). This factorization has shown promise in simultaneously improving positional fidelity and generative sharpness.

Despite these advances, key limitations persist for hybrid architectures that restrict their scalability and generalization. For example, CasCast (Gong et al., 2024) relies on a latent-space formulation,

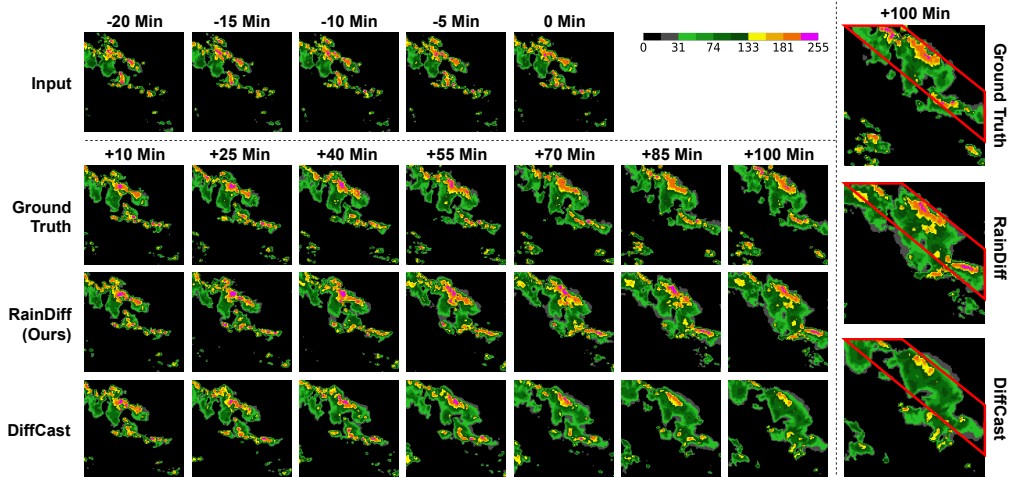

Figure 1: A visualization from the SEVIR dataset shows that, at the longest forecast horizon, Rain-Diff avoids oversmoothed outputs and better preserves weather fronts compared to the state-of-the-art DiffCast (Yu et al., 2024), resulting in closer alignment with the ground truth.

requiring an auxiliary autoencoder pre-trained on large datasets. This dependency hampers generalization to new domains where suitable autoencoders may be unavailable. In contrast, DiffCast (Yu et al., 2024) operates directly in pixel space, thereby avoiding latent bottlenecks. However, to remain computationally tractable, it omits self-attention mechanisms (Dosovitskiy et al., 2021) in the high-resolution layers. This design choice limits the model's capacity to capture complex long-range and fine-scale spatio-temporal dependencies, as illustrated in Figure 1.

To overcome these limitations, we propose Token-wise Attention, integrated across all spatial resolutions in our network. This design enables accurate modeling of fine-scale structures while maintaining computational efficiency. Unlike conventional self-attention (Yu et al., 2024; Gong et al., 2024), our token-wise formulation avoids the quadratic complexity induced by the high dimensionality of radar data. Moreover, all operations are performed directly in pixel space, removing the need for an external latent autoencoder (Gong et al., 2024). Finally, drawing on empirical insights, we introduce Post-attention, which emphasizes the informative conditional context crucial for the denoising process. To summarize, our key contributions are:

- We introduce Token-wise Attention, a novel mechanism that enables full-resolution self-attention directly in pixel space while retaining tractable computational cost.
- We provide a theoretical analysis exposing the limitations of integrating existing attention mechanisms into recurrent conditioners, and introduce Post-attention, a lightweight drop-in module that extracts critical contextual information to guide denoising while maintaining computational efficiency.
- We perform extensive experiments on four benchmark datasets, demonstrating that our approach consistently outperforms state-of-the-art methods in both deterministic and generative settings, achieving superior performance across multiple evaluation metrics.

## 2 RELATED WORK

Recently, deep learning has emerged as a powerful alternative to traditional Numerical Weather Prediction (NWP) (Skamarock et al., 2008), with approaches typically classified as deterministic or probabilistic. Early efforts were predominantly deterministic, emphasizing spatio-temporal modeling to produce point forecasts of future atmospheric conditions. For example, ConvLSTM (Shi et al., 2015) combined convolutional and recurrent layers to capture spatio-temporal dynamics. Later methods sought to enhance accuracy by integrating physical constraints, as in PhyDNet (Guen & Thome, 2020), or by incorporating a broader set of meteorological variables for more comprehensive forecasting, as in Pangu (Bi et al., 2023) and Fengwu (Chen et al., 2023). Although

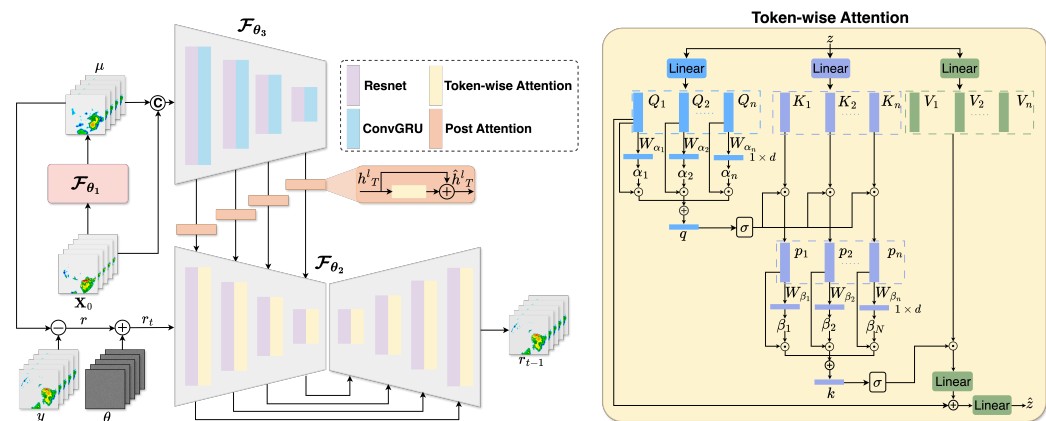

Figure 2: Overall architecture of our precipitation nowcasting framework RainDiff. Given an input sequence $X_0$, a deterministic predictor $\mathcal{F}_{\theta_1}$ outputs a coarse prediction $\mu$. The concatenation of $X_0$ and $\mu$ is encoded by a cascaded spatio-temporal encoder $\mathcal{F}_{\theta_3}$ to yield conditioning features $h$, refined by Post-attention. A diffusion-based stochastic module $\mathcal{F}_{\theta_2}$ equipped with Token-wise Attention at all resolutions in pixel space predicts residual segments $\hat{r}$ autoregressively, where the denoising process is conditioned on $h$ and the predicted segments. This design captures rich contextual relationships and inter-frame dependencies in the radar field while keeping computation efficient.

these models effectively capture large-scale motion patterns, their predictions tend to become overly smooth and blurry at longer lead times. This degradation arises from compounding errors, reliance on Mean Squared Error (MSE) loss, and the absence of local stochastic modeling—all of which suppress fine-scale detail.

Generative models have been proposed to mitigate the blurriness of deterministic forecasts by introducing latent variables that capture the inherent stochasticity of weather patterns. Examples include GAN-based approaches such as DGMR (Ravuri et al., 2021) and more recently, diffusion-based models like PreDiff (Gao et al., 2023). While some generative methods treat the entire system stochastically, a growing line of research explores hybrid strategies. Notably, DiffCast (Yu et al., 2024) and CasCast (Gong et al., 2024) combine deterministic modeling of large-scale motion with probabilistic modeling of fine-scale variability, thereby leveraging the complementary strengths of both paradigms.

Although diffusion-based methods have achieved promising results, they often face limitations such as the training overhead of external latent autoencoders or the omission of attention layers due to the high computational cost of operating in pixel space. These trade-offs between representational capacity and computational feasibility are not unique to weather forecasting, but are symptomatic of diffusion models more broadly, including in domains such as medical imaging, where pretrained autoencoders are frequently unavailable (Chen et al., 2024; Konz et al., 2024). To overcome these limitations, we propose a method that simplifies the training pipeline and improves generality, enabling wide applicability without reliance on domain-specific latent autoencoders.

## 3 METHODOLOGY

We formulate precipitation nowcasting as a spatio-temporal forecasting problem on a hybrid framework, which consists of three components: a deterministic module $\mathcal{F}_{\theta_1}(\cdot)$, a diffusion-based stochastic module $\mathcal{F}_{\theta_2}(\cdot)$, and a spatio-temporal module $\mathcal{F}_{\theta_3}(\cdot)$. The output from $\mathcal{F}_{\theta_1}(\cdot)$ is passed to $\mathcal{F}_{\theta_3}(\cdot)$ to extract conditioning features, which guide the denoising process of $\mathcal{F}_{\theta_2}(\cdot)$.

We also introduce Token-wise Attention, a mechanism that enables self-attention at all spatial resolutions while avoiding the prohibitive pixel-space cost of Vision Transformer (ViT) self-attention (Dosovitskiy et al., 2021) under equivalent resource constraints. In contrast, prior approaches either use external latent autoencoders that compress inputs before training a U-Net from $\mathcal{F}_{\theta_2}(\cdot)$ and decode outputs during inference, or they restrict ViT self-attention to bottleneck resolutions because

of its high computational cost, especially the softmax operation on the attention map. In addition, we propose Post-attention in the spatio-temporal encoder $\mathcal{F}_{\theta_3}(\cdot)$, which emphasizes informative contextual signals to guide the denoising process.

## 3.1 OVERALL FRAMEWORK

Let $X_0 \in \mathbb{R}^{H \times W \times C \times T_{\text{in}}}$ be a 4-dimensional tensor of shape $(H, W, C, T_{\text{in}})$, representing a sequence of $T_{\text{in}}$ input frames, where $H$ and $W$ denote the spatial resolution and $C$ the number of channels. Similarly, let $y \in \mathbb{R}^{H \times W \times C \times T_{\text{out}}}$ denote the sequence of $T_{\text{out}}$ future frames. Our objective is to learn a generative model for the conditional distribution $p(y \mid X_0)$.

Our approach proceeds in two steps. First, we train a deterministic predictor $\mathcal{F}_{\theta_1} : \mathbb{R}^{H \times W \times C \times T_{\text{in}}} \rightarrow \mathbb{R}^{H \times W \times C \times T_{\text{out}}}$, to estimate the conditional expectation $\mu(X_0) = \mathbb{E}[y \mid X_0]$. This estimate provides only a coarse estimate of the conditional distribution, capturing the global motion trend and overall structure but failing to represent uncertainty and often leading to blurry predictions with a loss of fine-scale details. Second, we introduce a spatio-temporal encoder $\mathcal{F}_{\theta_3}(\cdot)$ that processes both $X_0$ and $\mu$ to extract a representation $h$, which encodes global motion priors, sequence consistency, and inter-frame dependencies. We then model the residual $r = y - \mu$, using a stochastic prediction module $\mathcal{F}_{\theta_2}(\cdot)$ based on a diffusion model (Section 3.2). Token-wise Attention (Section 3.3) refines the temporal evolution of the residual distribution conditioned on $h$, while Post-attention mechanism (Section 3.4) sharpens $h$ during denoising, amplifying salient context and suppressing irrelevant detail.

At inference, to generate a sample from $p(y \mid X_0)$, we first sample a residual $\hat{r}$ from the diffusion-based prediction module and then add it to the predicted mean $\hat{\mu}$, yielding one realization $\hat{y} = \hat{\mu} + \hat{r}$. Repeating this procedure produces diverse realizations of plausible future sequences. An overview of the proposed framework is shown in Figure 2.

## 3.2 STOCHASTIC MODELING NETWORK

Given a tensor of input frames $X_0$, the deterministic predictor $\mathcal{F}_{\theta_1}(\cdot)$ estimates the conditional mean $\mu(X_0)$ by minimizing the MSE loss:

$$\mathcal{L}(\theta_1) = \mathbb{E}\left[\|\mathcal{F}_{\theta_1}(X_0) - y\|^2\right]. \tag{1}$$

While $\mathcal{F}_{\theta_1}$ provides a deterministic estimate of the mean, such forecasts often blur intense echoes and lose fine-scale structure at long lead times. To address this, we incorporate a diffusion-based stochastic module $\mathcal{F}_{\theta_2}(\cdot)$, which learns a generative model for the conditional distribution $p(y \mid X_0)$ by iteratively denoising toward the data manifold (Ho et al., 2020; Song et al., 2020). We denote the resulting distribution as $p_{\theta_2}(y \mid X_0)$.

In the radar-echo domain, CorrDiff (Mardani et al., 2025) highlights a strong input–target distribution mismatch caused by large forward noise. This mismatch becomes more pronounced at longer horizons when diffusing directly on $y$, ultimately reducing sample fidelity. To mitigate this, we instead model the residual $r$, which lowers variance and enables learning $p_{\theta_2}(r \mid X_0)$ more effectively than $p_{\theta_2}(y \mid X_0)$ (Mardani et al., 2025).

Furthermore, we introduce a spatio-temporal encoder $\mathcal{F}_{\theta_3}(\cdot)$, which takes $(X_0, \mu)$ as input and produces a global feature representation:

$$h = \mathcal{F}_{\theta_3}\left(\text{cat}(X_0, \mu)\right), \tag{2}$$

which provides a compact summary of the temporal dynamics and captures the overarching motion trends and overall structure.

In particular, we model the residual sequence in an autoregressive factorization conditioned on the global representation $h$. The joint distribution over the residuals is expressed as:

$$p_{\theta_2}\left(r_{1:T_{\text{out}}} \mid h\right) = \prod_{j=1}^{T_{\text{out}}} p_{\theta_2}\left(r_j \mid r_{j-1}, h\right). \tag{3}$$

Recent work (Ning et al., 2023) has shown that sequence-to-sequence multi-horizon forecasting provides a more effective paradigm than one-step autoregressive prediction for recurrent spatio-temporal modeling. This strategy mitigates error accumulation, improves temporal coherence,

and enhances computational efficiency. Motivated by these benefits, we partition the residual sequence $r$ into contiguous segments, defining $s_m = r_{[(m-1)T_{in}:mT_{in}]}, \quad m = 1, \ldots, M$ where $M = \left\lceil \frac{T_{out}}{T_{in}} \right\rceil$ and each $s_m \in \mathbb{R}^{H \times W \times C \times M}$. The full sequence is then obtained by concatenation: $s = \text{cat}(s_1, s_2, \ldots, s_M) \in \mathbb{R}^{H \times W \times C \times M \times T_{out}}$.

We model the evolution of the segment sequence $s$ in an autoregressive manner using a backward diffusion process: each segment $s_m$ is predicted conditioned on its predecessor $s_{m-1}$ and the global context $h$. The joint distribution is expressed as:

$$p_{\theta_2}(s_{1:M} \mid h) = \prod_{m=1}^{M} p_{\theta_2}(s_m \mid s_{m-1}, h). \tag{4}$$

During inference, since the ground-truth $s_{m-1}$ is not available, it is replaced by the estimated $\hat{s}_{m-1}$ generated at step $(m-1)$.

Diffusion models (Song et al., 2020; Ho et al., 2020) consist of a fixed forward noising process and a learned reverse denoising process. In the forward chain, starting from a clean segment $s_m^0 \sim p(s)$ (with $s_m^0 \equiv s_m$), Gaussian noise is added according to a variance schedule $\{(\alpha_t, \beta_t)\}_{t=1}^{\mathcal{T}}$, where $\beta_t = 1 - \alpha_t$: $q(s_m^t \mid s_m^{t-1}) = \mathcal{N}(s_m^t; \sqrt{\alpha_t} s_m^{t-1}, \beta_t I)$. This admits a closed-form expression for sampling at any step $t \in \{1, 2, \ldots, \mathcal{T}\}$: $q(s_m^t \mid s_m^0) = \mathcal{N}(s_m^t; \sqrt{\bar{\alpha}_t} s_m^0, (1 - \bar{\alpha}_t)I)$, with $\bar{\alpha}_t = \prod_{k=1}^{t} \alpha_k$. After $\mathcal{T}$ steps, $s_m^{\mathcal{T}}$ approaches standard Gaussian noise. The reverse process is learned as $p_{\theta_2}(s_m^{t-1} \mid s_m^t, \hat{s}_{m-1}, h)$, which iteratively denoises $s_m^t$ toward the data manifold conditioned on the previously predicted segment $\hat{s}_{m-1}$ and global context $h$. For a $\mathcal{T}$-step denoising diffusion, the target distribution is modeled as:

$$p_{\theta_2}(s_m^{0:\mathcal{T}} \mid \hat{s}_{m-1}, h) = p(s_m^{\mathcal{T}}) \prod_{t=1}^{\mathcal{T}} p_{\theta_2}(s_m^{t-1} \mid s_m^t, \hat{s}_{m-1}, h). \tag{5}$$

where $s_m^{\mathcal{T}} \sim \mathcal{N}(0, I)$, $t$ indexes the denoising step, and $s_m^t$ denotes the $t$-th denoising state of the $m$-th residual segment.

In the denoising process, learning to recover the residual state $s_m^{t-1}$ from $s_m^t$ is equivalent to estimating the noise $\epsilon$ injected at the $t$-th corruption step. Accordingly, the diffusion module $\mathcal{F}_{\theta_2}(\cdot)$ is trained with the segment-level loss:

$$\mathcal{L}(\theta_2, \theta_3; s_m) = \mathbb{E}\left[\left\|\mathcal{F}_{\theta_2}(s_m^t; \hat{s}_{m-1}, h, t, m) - \epsilon\right\|^2\right]. \tag{6}$$

where $\theta_3$ are the parameters for the global representation $h$. The overall diffusion loss is then obtained by aggregating over all residual segments:

$$\mathcal{L}(\theta_2, \theta_3) = \sum_{m=1}^{M} \mathcal{L}(\theta_2, \theta_3; s_m). \tag{7}$$

Finally, to capture the interaction between the deterministic predictive backbone and the stochastic residual diffusion, we train the entire framework end-to-end with the combined objective:

$$\mathcal{L}(\theta_1, \theta_2, \theta_3) = \gamma \mathcal{L}(\theta_2, \theta_3) + (1 - \gamma)\mathcal{L}(\theta_1). \tag{8}$$

where $\gamma \in [0, 1]$ balances the contributions of the stochastic and deterministic components.

Once trained, the diffusion module generates each residual segment $s_m$ by iteratively denoising from Gaussian noise, conditioned on the previously predicted segment $\hat{s}_{m-1}$. Repeating this procedure for $M$ steps yields a residual sequence $\hat{s} \in \mathbb{R}^{H \times W \times C \times M \times T_{out}}$, with the initial segment $\hat{s}_0$ initialized to zeros. A single realization of the future sequence is then obtained by adding the sampled residual $\hat{s}$ to the deterministic mean $\hat{\mu}$: $\hat{y} = \hat{\mu} + \hat{s}$. Repeating the sampling procedure produces multiple realizations drawn from the learned approximate conditional distribution $p(y \mid h)$.

### 3.3 TOKEN-WISE ATTENTION

Recent studies (Shaker et al., 2023) have simplified attention mechanisms by discarding key–value interactions and retaining only query–key interactions to model token dependencies. However, our

empirical analysis shows that relying solely on query–key interactions fails to capture the detailed characteristics of radar echoes, as it ignores the contribution of value (V) information. To overcome this limitation, we introduce *Token-wise Attention* (TWA).

Given an input embedding matrix $z \in \mathbb{R}^{n \times d}$, where $n$ denotes the number of tokens and $d$ the embedding dimension, the self-attention mechanism in ViT (Dosovitskiy et al., 2021) has a computational complexity of $O(n^2 d)$. In contrast, our Token-wise Attention achieves a reduced complexity of $O(nd)$. For a feature map of spatial size $h \times w$ (i.e., $n = h \times w$), this reduction translates from $O(h^2 w^2 d)$ to $O(hwd)$.

First, the input $z$ is projected into query, key, and value representations via linear transformations: $Q = zW_Q, \quad K = zW_K, \quad V = zW_V$, where $W_Q, W_K, W_V \in \mathbb{R}^{d \times d}$. Each matrix can then be expressed as

$$Q = [q_1, q_2, \ldots, q_n], \quad K = [k_1, k_2, \ldots, k_n], \quad V = [v_1, v_2, \ldots, v_n],$$

with $q_i, k_i, v_i \in \mathbb{R}^{1 \times d}$.

To highlight the token-wise importance within the sequence $Q$, we introduce a learnable weight vector $w_\alpha \in \mathbb{R}^{1 \times d}$. This vector interacts with the query matrix $Q \in \mathbb{R}^{n \times d}$ through a scaled dot product, yielding a query score map $\alpha \in \mathbb{R}^{1 \times n}$. The entries of $\alpha$ represent attention weights that quantify the relative significance of each query token $q_i \in \mathbb{R}^{1 \times d}$ with respect to the global context defined by $w_\alpha$. These weights are then used to construct a global query representation $q \in \mathbb{R}^{1 \times d}$ by aggregating information across all tokens. Specifically, we compute a normalized weighted sum of the query tokens via a Softmax function:

$$q = \text{Softmax}\left(\sum_{i=1}^{n} \alpha_i q_i\right), \quad \alpha = Q \cdot \frac{w_\alpha}{\sqrt{d}}. \tag{9}$$

Unlike ViT self-attention, which applies a Softmax over an $n \times n$ similarity matrix, our approach normalizes only along a $1 \times n$ dimension. The resulting global query $q$ aggregates information from all token-level queries, emphasizing components with greater attention relevance as determined by the learned distribution $\alpha$. Subsequently, the global query $q$ is compared against each key token $k_i \in \mathbb{R}^{1 \times d}$ from the key matrix $K \in \mathbb{R}^{n \times d}$. This comparison is computed via dot products, yielding the query–key alignment matrix $p \in \mathbb{R}^{n \times d}$:

$$p = [p_1, p_2, \ldots, p_n] = [q \cdot k_1, \ q \cdot k_2, \ \ldots, \ q \cdot k_n]. \tag{10}$$

Similar to equation 9, we summarize the global key $k \in \mathbb{R}^{1 \times d}$ as:

$$k = \text{Softmax}\left(\sum_{i=1}^{n} \beta_i p_i\right), \quad \beta = K \cdot \frac{w_\beta}{\sqrt{d}}. \tag{11}$$

Finally, the interaction between the global key vector $k \in \mathbb{R}^{1 \times d}$ and the value matrix $V \in \mathbb{R}^{n \times d}$ is modeled through element-wise multiplication, producing a global context representation. To refine the token representations, we apply two multilayer perceptrons (MLPs): one operating on the normalized queries with a residual connection, and the other on the key–value interaction. The resulting output $\hat{z}$ is expressed as:

$$\hat{z} = \text{MLP}_Q(\text{Norm}(Q)) + \text{MLP}_V(V * k) \tag{12}$$

where $*$ denotes element-wise multiplication.

### 3.4 SPATIO-TEMPORAL ENCODER

**Spatio-temporal encoder:** In our RainDiff framework, the spatio-temporal encoder $\mathcal{F}_{\theta_3}(\cdot)$ is built as a cascaded architecture to extract conditioning features at multiple resolutions. Specifically, $\mathcal{F}_{\theta_3}$ comprises several resolution-aligned blocks; each block contains a ResNet module $R^l$ for spatial feature extraction and a ConvGRU module $\text{G}^l$ for temporal modeling across $T_{\text{in}} + T_{\text{out}}$ frames:

$$h_j^l = \text{G}^l\big(R^l(x_j^l), \, h_{j-1}^l\big), \qquad j = 1, 2, \ldots, T_{\text{in}} + T_{\text{out}}. \tag{13}$$

Here, $x_j^l$ and $h_{j-1}^l$ denote the $j$-th input and $(j-1)$-th hidden state at level $l$, respectively. When $l=0$, $x_j^0$ is the raw input frame, and we hence can write $x_j \equiv x_j^0$.

**Post-attention (PA):** Due to the absence of a latent autoencoder, the conditioning produced by the spatio-temporal encoder can have redundant context. A self-attention mechanism is thus needed to suppress irrelevant content and emphasize salient context for conditioning the diffusion model. To do this, prior work often inserts attention inside recurrent modules (Lange et al., 2021; Lin et al., 2020). In our setting, however, the training objective does not directly supervise temporal recurrence; it is defined by denoising (diffusion) and deterministic reconstruction. In addition, as conditioning sequences are encoded one-by-one and gradients propagate to the spatio-temporal encoder through two pathways (via $h$ and via $\mu$), the resulting gradient with respect to each input frame $x_j$ decomposes as:

$$\frac{\partial \mathcal{L}_{123}}{\partial x_j} = \gamma \frac{\partial \mathcal{L}_{23}}{\partial h_T} \frac{\partial h_T}{\partial x_j} + \left[\gamma\left(\frac{\partial \mathcal{L}_{23}}{\partial h_T} \frac{\partial h_T}{\partial \mu} - \sum_{m,t} \sqrt{\bar{\alpha}_t} \frac{\partial \mathcal{L}_{23}}{\partial s_m^t}\right) + (1-\gamma)\frac{\partial \mathcal{L}_1}{\partial \mu}\right]\frac{\partial \mu}{\partial x_j} \quad (14)$$

$$\frac{\partial h_T}{\partial x_j} = \left(\prod_{i=j}^{T-1} \frac{\partial h_{i+1}}{\partial h_i}\right)\frac{\partial h_j}{\partial x_j}, \frac{\partial h_T}{\partial \mu} = \left(\prod_{i=1}^{T-1} \frac{\partial h_{i+1}}{\partial h_i}\right)\frac{\partial h_1}{\partial \mu}, T = T_{\text{in}} + T_{\text{out}}, \ \ j \in \{1, \ldots, T_{\text{in}}\}.$$
$$(15)$$

where $\mathcal{L}_{123}, \mathcal{L}_{23}, \mathcal{L}_1$ denote $\mathcal{L}(\theta_1, \theta_2, \theta_3), \mathcal{L}(\theta_2, \theta_3), \mathcal{L}(\theta_1)$ respectively. In spatio-temporal encoders, gradients can suffer from severe attenuation due to repeated multiplication of Jacobians, i.e., through the product $\prod_{i=j}^{T-1} \partial h_{i+1}/\partial h_i$. Inserting attention within each recurrent step adds an extra per-step contraction and ties the attention update to intermediate gradients that are not directly anchored to the dedicated loss, which worsens attenuation. We therefore apply our Token-wise Attention after the encoder outputs (PA), at multiple resolutions. Post-attention brings two practical advantages: (i) fewer attention calls—attention is applied once per encoded representation (per scale), rather than at every recurrent step, substantially reducing compute versus in-block attention (Lange et al., 2021; Lin et al., 2020); and (ii) stability and efficiency—by avoiding attention inside recurrence, PA reduces gradient attenuation and amplification through long products and improves training stability and throughput. Further experiments in the ablation studies (Section 4.4) support these viewpoints.

# 4 EXPERIMENTS

## 4.1 IMPLEMENTATION DETAILS

**Dataset:** We evaluate our framework on four widely used precipitation nowcasting datasets Shanghai Radar (Chen et al., 2020), SEVIR (Veillette et al., 2020), MeteoNet (Larvor et al., 2020) and CIKM[1]. We adopt a challenging forecasting setup of predicting 20 future frames from 5 initial frames ($5 \rightarrow 20$), except for the CIKM dataset, where only the next 10 frames are predicted ($5 \rightarrow 10$) due to its shorter sequence length constraints. Further dataset details are provided in Appendix B.

**Training protocol:** Our RainDiff model is trained for 300K iterations on a batch size of 4 via an Adam optimizer with a learning rate of $1 \times 10^{-4}$. Following (Ho et al., 2020), we set the diffusion process to 1000 steps and employ 250 denoising steps during inference using DDIM (Song et al., 2020). We implement SimVP (Gao et al., 2022a) as our deterministic module. In line with (Yu et al., 2024), the combined loss in Equation 8 is balanced with $\gamma = 0.5$ between deterministic and denoising components. All experiments are executed on a single NVIDIA A6000 GPU.

## 4.2 EVALUATION METRICS

Forecast accuracy is evaluated using average Critical Success Index (CSI) and Heidke Skill Score (HSS) across multiple reflectivity thresholds (Luo et al., 2022; Gao et al., 2023; Veillette et al., 2020). To assess spatial robustness, we also report multi-scale CSI scores (CSI-4, CSI-16) using pooling kernels of size 4 and 16 (Gao et al., 2022b; 2023). Perceptual quality is quantified by Learned Perceptual Image Patch Similarity (LPIPS) and Structural Similarity Index Measure (SSIM).

---

[1] https://tianchi.aliyun.com/dataset/1085

Table 1: Quantitative comparison across four radar nowcasting datasets (Shanghai Radar, MeteoNet, SEVIR, CIKM). We evaluate deterministic baselines (PhyDNet, SimVP, EarthFarseer, AlphaPre) and probabilistic methods (DiffCast) against our RainDiff using CSI, pooled CSI at $4\times4$ and $16\times16$ (CSI-4 / CSI-16), HSS, LPIPS, and SSIM. **Bold** marks our results. Overall, RainDiff attains the best or tied-best performance on most metrics and datasets, indicating both stronger localization and perceptual/structural quality. This design allows capturing rich context and dependency between frames in the radar field while maintaining efficient computation.

| Method | Shanghai Radar | | | | | | MeteoNet | | | | | |
|---|---|---|---|---|---|---|---|---|---|---|---|---|
| | ↑CSI | ↑CSI-4 | ↑CSI-16 | ↑HSS | ↓LPIPS | ↑SSIM | ↑CSI | ↑CSI-4 | ↑CSI-16 | ↑HSS | ↓LPIPS | ↑SSIM |
| PhyDNet | 0.3692 | 0.4066 | 0.5041 | 0.5009 | 0.2505 | 0.7784 | 0.1259 | 0.1450 | 0.1741 | 0.1950 | 0.2837 | 0.8188 |
| SimVP | 0.3965 | 0.4360 | 0.5261 | 0.5290 | 0.2365 | 0.7727 | 0.1300 | 0.1662 | 0.2190 | 0.1927 | 0.2448 | 0.8098 |
| EarthFarseer | 0.3998 | 0.4455 | 0.5405 | 0.5330 | 0.2126 | 0.7214 | 0.1651 | 0.2230 | 0.3567 | 0.2527 | 0.2128 | 0.7548 |
| DiffCast | 0.4000 | 0.4887 | 0.6063 | 0.5358 | 0.1561 | 0.7898 | 0.1454 | 0.2209 | 0.3382 | 0.2196 | 0.1298 | 0.7923 |
| AlphaPre | 0.3934 | 0.3939 | 0.4237 | 0.5203 | 0.2925 | 0.7863 | 0.1532 | 0.1729 | 0.1965 | 0.2284 | 0.2697 | 0.7891 |
| **RainDiff** | **0.4448** | **0.5152** | **0.6260** | **0.5822** | **0.1454** | **0.7997** | **0.1618** | **0.2484** | **0.3907** | **0.2430** | **0.1231** | **0.8210** |

| Method | SEVIR | | | | | | CIKM | | | | | |
|---|---|---|---|---|---|---|---|---|---|---|---|---|
| | ↑CSI | ↑CSI-4 | ↑CSI-16 | ↑HSS | ↓LPIPS | ↑SSIM | ↑CSI | ↑CSI-4 | ↑CSI-16 | ↑HSS | ↓LPIPS | ↑SSIM |
| PhyDNet | 0.3648 | 0.3878 | 0.4618 | 0.4400 | 0.4057 | 0.5606 | 0.4487 | 0.4790 | 0.5488 | 0.4906 | 0.5079 | 0.4906 |
| SimVP | 0.3572 | 0.3766 | 0.4229 | 0.4268 | 0.4604 | 0.4898 | 0.4879 | 0.5079 | 0.5817 | 0.5328 | 0.5574 | 0.5272 |
| EarthFarseer | 0.3677 | 0.4120 | 0.5310 | 0.4459 | 0.3124 | 0.5264 | 0.4647 | 0.4819 | 0.5651 | 0.5094 | 0.4960 | 0.5572 |
| DiffCast | 0.3711 | 0.4417 | 0.6168 | 0.4539 | 0.2137 | 0.5362 | 0.4834 | 0.5175 | 0.6481 | 0.5182 | 0.2900 | 0.4993 |
| AlphaPre | 0.3436 | 0.3578 | 0.4010 | 0.4038 | 0.4005 | 0.5452 | 0.4858 | 0.5101 | 0.6064 | 0.5231 | 0.4660 | 0.4852 |
| **RainDiff** | **0.3835** | **0.4534** | **0.6193** | **0.4701** | **0.2070** | **0.5500** | **0.4916** | **0.5235** | **0.6536** | **0.5236** | **0.2926** | **0.5110** |

## 4.3 EXPERIMENTAL RESULTS

For a comprehensive evaluation, we compare our method against both deterministic and probabilistic baselines. The deterministic models include the recurrent-free SimVP (Gao et al., 2022a) and AlphaPre (Lin et al., 2025), as well as the autoregressive PhyDNet (Guen & Thome, 2020) and EarthFarseer (Wu et al., 2024). As the state-of-the-art probabilistic approach, we include the DiffCast (Yu et al., 2024) model.

**Quantitative results:** Table 1 presents the results of our RainDiff compared to other baselines across four radar datasets. On the Shanghai Radar dataset, RainDiff achieves the highest CSI (0.4448), HSS (0.5822), and SSIM (0.7997), along with the lowest LPIPS (0.1454), significantly outperforming the next-best method, DiffCast, across all metrics. Similarly, on SEVIR dataset, RainDiff achieves the best CSI (0.3835), LPIPS (0.2070), and competitive SSIM (0.5500), offering a better perceptual trade-off than PhyDNet, which has higher SSIM (0.5606) but much worse LPIPS (0.4057). For CIKM dataset, RainDiff leads with the best CSI (0.4916), CSI-4 (0.5235), and CSI-16 (0.6536), demonstrating strong robustness under high-variability conditions. On MeteoNet, RainDiff delivers the best perceptual scores (SSIM: 0.8201, LPIPS: 0.1231) and ranks second in CSI (0.1618), confirming its strong generalization. In addition, Figure 4 reports frame-wise CSI and HSS. As lead time increases, scores drop across all methods due to accumulating forecast uncertainty, yet our approach consistently outperforms the baselines at most timesteps—often by a larger margin at longer leads—demonstrating superior robustness to temporal expanding.

**Qualitative results:** A comparison in Figure 3 reveals the limitations of existing methods. Deterministic models yield blurry outputs, while the stochastic model DiffCast, though sharper, introduces excessive and uncontrolled randomness at air masses' boundaries—an issue we attribute to its lack of attention mechanisms. This results in an unstable representation of temporal-spatial dependencies. Our framework solves this by integrating Token-wise Attention. This component not only enables the generation of realistic, high-fidelity details but also regulates the model's stochastic behavior, leading to forecasts with improved structural accuracy and consistency, thereby mitigating the chaotic predictions seen in DiffCast. Further visualizations are given in Appendix E.

## 4.4 ABLATION STUDY

**Effect of Individual Components:** To evaluate the contribution of each component, we perform ablation experiments with three settings: (i) RainDiff without both Token-wise Attention in the U-

Table 2: Ablation studies of RainDiff on the Shanghai Radar dataset: (a) individual components and (b) attention mechanisms in the spatio-temporal encoder.

(a) Ablation: individual components (i–iv).

| Method | ↑CSI | ↑CSI-4 | ↑CSI-16 | ↑HSS | ↓LPIPS | ↑SSIM |
|--------|------|--------|---------|------|--------|-------|
| (i) | 0.4000 | 0.4887 | 0.6063 | 0.5358 | 0.1561 | 0.7898 |
| (ii) | 0.4370 | 0.5026 | 0.6030 | 0.5737 | 0.1461 | 0.7890 |
| (iii) | 0.4396 | 0.5066 | 0.6142 | 0.5767 | 0.1466 | 0.8125 |
| (iv) | 0.4448 | 0.5152 | 0.6260 | 0.5822 | 0.1454 | 0.7997 |

(b) Ablation: attention mechanisms (i–iii) on the spatio-temporal encoder.

| Method | ↑CSI | ↑CSI-4 | ↑CSI-16 | ↑HSS | ↓LPIPS | ↑SSIM |
|--------|------|--------|---------|------|--------|-------|
| (i) | 0.4284 | 0.4808 | 0.5600 | 0.5623 | 0.1562 | 0.8217 |
| (ii) | 0.4310 | 0.5060 | 0.6049 | 0.5680 | 0.1502 | 0.7751 |
| (iii) | 0.4448 | 0.5152 | 0.6260 | 0.5822 | 0.1454 | 0.7997 |

Figure 3: Qualitative comparison with existing works on the Shanghai Radar dataset, where the reflectivity range is on the top right.

Figure 4: Frame-wise CSI and HSS for various methods on the Shanghai Radar dataset.

Net and Post-attention in the Spatio-temporal Encoder, which corresponds to DiffCast (Yu et al., 2024), (ii) Integrate DiffCast with Adaptive Attention from (Shaker et al., 2023), (iii) RainDiff with Token-wise Attention and without Post-attention, and (iv) our full RainDiff model. As shown in Table 2a, the absence of any component results in a clear degradation of performance, underscoring the critical role of each design choice in strengthening predictive capability.

**Effect of attention mechanism on Spatio-temporal encoder:** As shown in Table 2b, we evaluate the effectiveness of our attention design on spatio-temporal encoder by comparing it with several alternatives proposed in (Lange et al., 2021; Lin et al., 2020), where attention layers are integrated within the recurrent block. We perform ablation experiments with three settings on ConvGRU block $l$-th: (i) The attention layers are integrated to the input $x_j^l$ at each frame $j$-th, (ii) The attention layers are integrated to the output $h_j^l$ at each frame $j$-th, and (iii) our RainDiff, where Post-attention is only applied on the final condition $h_T^l$ of frame $T$-th, Section 3.4. The results in Table 2b support our contribution discussed in Section 3.4: RainDiff with Post-attention consistently achieves higher efficiency while maintaining performance comparable to other integration methods.

## 5 CONCLUSION

RainDiff is an end-to-end diffusion framework for precipitation nowcasting that applies Token-wise Attention at all spatial scales within a diffusion U-Net, eliminating the need for a latent autoencoder and improving scalability and performance. In addition, we propose Post-attention module mitigating gradient attenuation when attention meets recurrent conditioning. Across four benchmarks, RainDiff surpasses deterministic and probabilistic baselines in localization, perceptual quality, and long-horizon robustness. Future work will involve physical constraints by using multi-modal inputs and reduce latency by replacing autoregression.

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
