# OpenReview forum: "RainDiff: End to End Precipitation Nowcasting Via Token-wise Attention Diffusion"
_ICLR.cc/2026/Conference — ICLR 2026 Conference Withdrawn Submission_

### Official Review · Reviewer_Urr3 · 2025-10-25

**Soundness:** 3
**Presentation:** 2
**Contribution:** 2
**Rating:** 4
**Confidence:** 3

**Summary:**

This paper presents RainDiff, an end-to-end diffusion-based framework for precipitation nowcasting that directly models radar sequences in the pixel space without using latent autoencoders. To address the computational bottleneck of full-resolution attention, the authors propose a Token-wise Attention (TWA) mechanism that achieves global dependency modeling with linear complexity. In addition, a Post-attention (PA) module is introduced after the spatio-temporal encoder to mitigate gradient vanishing and enhance long-term temporal consistency. Experiments conducted on four radar datasets demonstrate that RainDiff achieves stable and competitive performance compared to existing deterministic and diffusion-based baselines, improving both forecast accuracy and spatial coherence in long-horizon precipitation prediction.

**Strengths:**

+ Clear motivation and well-defined goal. The paper clearly identifies the computational and architectural limitations of previous diffusion-based nowcasting models, and motivates the need for a fully end-to-end framework operating in pixel space.

+ Effective architectural design. The proposed Token-wise Attention (TWA) efficiently captures global spatial dependencies with linear complexity, making high-resolution attention feasible without latent compression. The introduction of the Post-attention (PA) module provides an effective way to stabilize gradients and improve temporal consistency in long-horizon predictions.

+ Comprehensive experiments. The paper includes extensive evaluations on four radar datasets with solid baselines, ablation studies, and visualization analyses, providing convincing empirical evidence of the method’s effectiveness.

**Weaknesses:**

+ Lack of theoretical depth. The paper introduces Token-wise Attention (TWA), but does not provide a rigorous mathematical analysis demonstrating that TWA is a simplified form of attention with equivalent representational properties. In its current form, TWA behaves more like a global aggregation mechanism than a true attention mechanism, as it lacks token-to-token interaction and selective dependency modeling.
+ Poor figure and visualization quality. The presentation of figures and tables in the paper is suboptimal and does not conform to standard academic conventions.  In Table 1, the authors highlight their own model in bold font, which is uncommon in scholarly publications.  The table layout also fails to clearly reflect the relative performance advantage of the proposed method.  In Table 6, the proposed RainDiff model is not included at all, resulting in an incomplete performance comparison. Several visualization figures contain too many elements within a single image, which affects the fluency of reading and comprehension.

**Questions:**

+ Could the authors provide a more rigorous theoretical justification for Token-wise Attention (TWA)?  Specifically, under what assumptions can TWA be viewed as a simplified or approximate form of standard self-attention, and what kinds of dependencies might be lost due to the global aggregation design?
+ How does TWA differ from previously proposed linear or low-rank attention mechanisms?  Is there any empirical or theoretical comparison that highlights its unique behavior?
+ While TWA reduces quadratic complexity, how does its runtime and memory usage compare with other linear attention models in practice?

---

### Official Review · Reviewer_rZXK · 2025-10-28

**Soundness:** 3
**Presentation:** 3
**Contribution:** 2
**Rating:** 2
**Confidence:** 4

**Summary:**

The paper proposes RainDiff, an end-to-end diffusion framework for precipitation nowcasting. It retains the deterministic–probabilistic hybrid design of DiffCast, combining a mean-field deterministic predictor with a residual diffusion model under a joint objective. The key claimed novelties are: (1) Token-Wise Attention (TWA) for linear-complexity global token aggregation, and (2) Post-Attention (PA) applied after the spatio-temporal encoder to enhance gradient stability. The method is evaluated on Shanghai, SEVIR, MeteoNet, and CIKM datasets and shows small but consistent improvements in CSI/HSS scores.

**Strengths:**

S1. Reproducible and consistent experimental setup across four datasets.
S2. Clear motivation to remove autoencoder dependency for cross-dataset generalization.
S3. Logical architecture design with good readability and structured methodology.
S4. Empirical improvements are directionally consistent and align with design goals.

**Weaknesses:**

W1. Limited methodological originality beyond architectural refinement of DiffCast.
W2. No theoretical or runtime analysis differentiating TWA from existing linear attentions (Performer, SwiftFormer).
W3. Efficiency claims (speed, memory) are unsupported.
W4. Qualitative and ablation studies are restricted to a single dataset.
W5. No statistical significance or confidence intervals for reported metrics.
W6. Figures and equations need better clarity and consistency

**Questions:**

1. Clarification is needed on how the proposed Token-Wise Attention mathematically differs from existing efficient-attention mechanisms in terms of formulation and information flow.
2. Quantitative evidence on FLOPs, memory usage, and inference speed would help verify the claimed computational efficiency of the new modules.
3. Since the reported CSI and HSS improvements are relatively small, statistical validation across multiple runs would strengthen the credibility of the results.
4. The claimed gradient-stability benefit of post-Attention would be more convincing with supporting analyses of gradient norms or training dynamics.
5. Visual and ablation results are shown only for the Shanghai dataset. Extending these evaluations to other datasets would demonstrate generalization.
6. A comparison against other efficient-attention variants implemented under the same setting would clarify whether the improvements arise from true methodological novelty.

---

### Official Review · Reviewer_3389 · 2025-10-30

**Soundness:** 3
**Presentation:** 3
**Contribution:** 3
**Rating:** 4
**Confidence:** 4

**Summary:**

This work proposes an end-to-end diffusion framework RainDiff for precipitation nowcasting, which integrates Token-wise Attention into U-Net and spatio-temporal encoder without latent autoencoders, and introduces Post-attention to optimize denoising. Experiments on 4 datasets show RainDiff outperforms baselines in various metrics results.

**Strengths:**

1. This work proposes Token-wise Attention, enabling full-resolution self-attention in pixel space, outperforming state-of-the-art baselines in multiple metrics on 4 datasets.
2. This work effectively designs a hybrid framework (deterministic + stochastic diffusion modules) and Post-attention to mitigate gradient attenuation.

**Weaknesses:**

1. The precipitation process is constrained by laws of atmospheric dynamics and thermodynamics, yet the hybrid framework proposed in this paper is entirely data-driven and does not incorporate any physical priors. Although this issue is mentioned in the limitations section, it may still lead to physically unreasonable results.
2. The paper only theoretically analyzes the optimization of time complexity, while lacking comparative analysis at the experimental level—such as the actual consumption of computing resources and time during the training and generation processes.
3. In precipitation nowcasting, the forecasting requirements and difficulty vary significantly across different lead times. For instance, short lead times focus on accurately capturing local details, while long lead times may prioritize stably modeling global dynamics. However, RainDiff lacks an analysis of the differences in its advantages under different lead times. For example, it is unclear whether the improvement of TWA on local details in short lead times is comparable to its effect on modeling global dependencies in long lead times.
4. As a probabilistic forecasting model, RainDiff lacks an analysis of the model's stability and its ability to control randomness (including handling outliers), .

**Questions:**

The questions here are related to the three points described as weaknesses.

---

### Official Review · Reviewer_AEhz · 2025-10-30

**Soundness:** 2
**Presentation:** 3
**Contribution:** 2
**Rating:** 6
**Confidence:** 5

**Summary:**

The paper proposes an end-to-end precipitation nowcasting/downscaling model that combines a deterministic predictor with a diffusion model operating on residuals. The key architectural element is a token-wise attention block that claims to reduce quadratic complexity while preserving long-range temporal dependencies. The method is evaluated on standard radar/climate datasets with metrics such as CSI/HSS (and perceptual scores), showing improvements over recent baselines.

**Strengths:**

- Quality/results: Solid quantitative gains on standard metrics (CSI/HSS and perceptual proxies). If robust, these are practically relevant.

- Clarity of task framing: Modeling residual stochasticity on top of a coarse predictor is appropriate for precipitation. This is consistent with prior residual video diffusion works.

- Engineering: The end-to-end pipeline appears implementable and computationally careful.

**Weaknesses:**

- No long-range dependency metric: Paper mentions lack of attn in Diffcast leads to long range dependency issues. However, DiffCast does have temporal attn, maybe a metric which explicitly captures long range dependency would have been ideal, something correlation based.

- Multi-loss design not justified: Prior residual diffusion works often use a single diffusion loss end-to-end, avoiding extra hyperparameters (like gamma). The deterministic modules naturally learn coarse features without having to rely on explicit loss. Single end-to-end diffusion loss is usually sufficient.

- Attention novelty/efficiency unclear. TWA reduces quadratic cost, but there is no head-to-head vs SwiftFormer (efficient additive), AFT (element-wise/global-summary), or linear/IO-aware variants (Linformer, Performer, Nyströmformer, FlashAttention). Wall-clock and memory comparisons seem to be missing.

- Limited probabilistic evaluation: Calibration/CRPS missing, usually needed to support uncertainty claims (common in diffusion-for-weather).

- Compute & latency not reported: No throughput/VRAM/sampling-step analysis relative to latent diffusion or efficient attention baselines.

**Questions:**

- Your text says DiffCast omits self-attention at high-resolution layers for tractability, not attention in general. Can you specify which attention forms you claim are absent in DiffCast (spatial vs temporal), and map RainDiff’s attention placement vs DiffCast layer-by-layer to make the distinction precise?

- Are tokens strictly per-pixel (n=h×w) at every scale, or do you patch/subsample anywhere? What positional encodings (if any) are used inside TWA at different resolutions

- You argue DiffCast produces uncontrolled randomness at air-mass boundaries. How did you quantify “boundary stability”?

---

### Note · Authors · 2025-11-14

**Comment:**

Dear Reviewers and Area Chairs, we have decided to withdraw our paper. We are very grateful for the time and expertise you invested in reviewing our submission.

**Withdrawal Confirmation:**

I have read and agree with the venue's withdrawal policy on behalf of myself and my co-authors.